# Mechanically flexible mid-wave infrared imagers using black phosphorus ink films

Theodorus Jonathan Wijaya [1,2,3,4,9], Naoki Higashitarumizu [1,2,3,9], Shifan Wang [1,2,3,5], Shogo Tajima [1,3,6], Hyong Min Kim [1,2,3], Shu Wang [2,3,7], Dehui Zhang [1,2,3], James Bullock [5], Tomoyuki Yokota [4] ✉, Takao Someya [4] ✉ & Ali Javey [1,2,3,7,8] ✉

The mid-wave infrared (MWIR) spectral range ($\lambda = 3$–$8\,\mu m$) enables important sensing and imaging applications, including non-invasive bioimaging, night vision, and autonomous navigation. Commercial MWIR photodetectors are limited to rigid imagers based on heteroepitaxial materials. There is an emerging need for mechanically flexible MWIR imagers to broaden their functionality and practicality. Recently, photodetectors using van der Waals (vdW) black phosphorus (BP) flakes have demonstrated highly sensitive room-temperature photodetection. Additionally, vdW materials are solution-processable, facilitating scalable processing and flexible device fabrication. In this work, we present flexible MWIR imagers consisting of photodiodes fabricated on thin plastic substrates using BP ink films. We demonstrate mechanically robust responsivity up to 2.5-mm bending radii and after 5000 bending cycles. Leveraging this flexibility, we achieve full-azimuthal imaging, detecting directional light sources with precision. These results establish a scalable approach for large-area, conformable MWIR imaging and pave the way for integration with flexible electronics.

Material innovations have played a pivotal role in advancing optoelectronics technologies across ultraviolet, visible, and near-infrared ranges. Notably, emerging commercial applications in these spectral ranges are driven by breakthroughs in organic materials, quantum dots, and perovskites[1–6]. For the mid-wave infrared (MWIR) range ($\lambda = 3$–$8\,\mu m$), commercial detectors have progressed by relying on thermopiles, bolometers, and photodiodes based on epitaxially grown HgCdTe, InSb, and InAs, that originated decades ago with minimal material advancements over the years[7–9]. As the demand for MWIR sensing and imaging continues to rapidly grow, especially in biomolecular sensing, gas detection, and night vision[10–13], advancements in

materials, fabrication processes, and device architectures are essential. In particular, the development of large-area, mechanically flexible imagers enables functionalities such as wide-angle detection, proximal sensing, surface-conformable imaging, and integration with mechanically deformable systems[14–16].

Recently, black phosphorus (BP), with a bandgap of 0.33 eV, has emerged as a promising material for MWIR photodetection. Mechanically exfoliated BP flakes have been integrated into photodiodes and photoconductors, demonstrating highly sensitive MWIR detection at room temperature[17–21]. To address the challenges associated with the uncontrolled isolation of BP flakes via tape exfoliation,

[1]Department of Electrical Engineering and Computer Sciences, University of California, Berkeley, CA, USA. [2]Materials Sciences Division, Lawrence Berkeley National Laboratory, Berkeley, CA, USA. [3]Berkeley Sensor & Actuator Center, University of California, Berkeley, CA, USA. [4]Department of Electrical Engineering and Information Systems, The University of Tokyo, 7-3-1 Hongo, Bunkyo-ku, Tokyo, Japan. [5]Department of Electrical and Electronic Engineering, University of Melbourne, Melbourne, VIC, Australia. [6]Azbil North America Research and Development, Inc, Santa Clara, CA, USA. [7]Department of Materials Science and Engineering, University of California, Berkeley, CA, USA. [8]Kavli Energy NanoScience Institute at the University of California, Berkeley, CA, USA. [9]These authors contributed equally: Theodorus Jonathan Wijaya, Naoki Higashitarumizu. ✉ e-mail: yokota@ntech.t.u-tokyo.ac.jp; someya@ee.t.u-tokyo.ac.jp; ajavey@berkeley.edu

solution-processed liquid-phase exfoliation offers a scalable alternative[22,23]. By formulating BP into an ink, centimeter-scale films have been fabricated, exhibiting MWIR emission and enabling proof-of-concept single-pixel MWIR detection under an applied bias[24]. Additionally, the ink-based BP film is compatible with various substrates, as its van der Waals (vdW) interfaces are free from lattice-matching constraints, enabling scalable fabrication on flexible polymer substrates. Moreover, sliding vdW junctions are expected to facilitate mechanical flexibility in the BP ink film by relieving local strain at the nanosheet interfaces[25,26].

However, demonstrating flexible MWIR imagers using BP ink films remains challenging due to the interplay of multiple factors. Thin BP films are essential for mechanical flexibility but must also maintain a pinhole-free structure across the entire imager area to prevent shunt pathways. In addition, smaller BP nanosheets enhance the uniformity of photodetector arrays, enabling high-yield imager pixel fabrication, whereas MWIR detection requires nanosheets thicker than ~6 nm (approximately 10 layers) to exhibit the desired optical bandgap of ~0.33 eV for bulk BP[27]. The thickness of liquid-phase-exfoliated nanosheets correlates with their lateral dimensions[28], further constraining the lower size limit.

In this study, we present mechanically flexible MWIR imagers by using BP ink films, demonstrating operation at room temperature without bias. To realize this flexible imager, the BP nanosheet dimensions were adjusted to achieve optimal MWIR detection, and the device architecture was designed to allow uniform coating of BP ink over a large area. The low-temperature processing of the ink film ($\leq$180 °C) enabled its scalable deposition onto flexible plastic substrates. These ink films were integrated into a heterostructure device architecture, incorporating interlayers that support uniformity and facilitate carrier separation. The photodiodes exhibited responsivities of up to tens of $\mu A \cdot W^{-1}$ across 3–4 $\mu m$ wavelength range at 0 V, a rise time of 4 $\mu s$, and a 3 dB cutoff frequency of 44 kHz. Using these photodetectors, a 6 × 6 array was fabricated to demonstrate imaging capabilities. The mechanical flexibility of the devices was validated by maintaining a stable photoresponse after bending test to a 2.5-mm radius and enduring over 5000 bending cycles with a 1-cm radius. Finally, by leveraging this flexibility, full-azimuthal panoramic imaging was achieved by expanding the field of view of a flexible imager, and a two-dimensional flexible scanner was realized.

## BP ink film for flexible MWIR imager

To reconcile the optical, electrical, and process engineering requirements of the BP ink film, this study utilized the probe sonication method for liquid-phase exfoliation of BP nanosheets from the bulk source. Probe sonication was selected over electrochemical intercalation, another established method for BP ink fabrication[29], for two primary reasons. First, probe sonication introduces fewer chemical contaminants by avoiding the use of salt-based ions, which may adhere strongly to exfoliated nanosheets. These residues, if not thoroughly removed, can adversely affect the conductivity and stability of the nanosheets. Second, probe sonication exhibits less variability than the electrochemical process owing to its simpler setup, resulting in greater reproducibility and better control over nanosheet dimensions. These advantages were found to be important for observing MWIR photodetection at 0 V and for achieving fabrication of large-scale imagers.

Subsequently, a two-step centrifugation process was introduced to reduce the variability of nanosheet thickness. This approach yielded nanosheets with an average lateral dimension of 416 ± 124 nm and an average thickness of 24 ± 15 nm, as determined from 300 nanosheets by using atomic force microscopy (AFM) on a diluted ink sample. The BP film formed with these dimensional characteristics exhibited MWIR detection across the imager area with a high yield, as elaborated below. Additional information on ink formulation is provided in Methods and

Supplementary Note 1. Other nanosheet dimensions investigated through different exfoliation settings are compared in Supplementary Table 1.

This BP film was incorporated into vertical photodetectors (illustrated in Fig. 1a), with adjacent layers of poly(3,4-ethylenedioxythiophene) polystyrene sulfonate (PEDOT:PSS) and 2,2′-((2Z,2′Z)-((12,13-bis(2-ethylhexyl)-3,9-diundecyl-12,13-dihydro[1,2,5]thiadiazolo[3,4-e]thieno[2,″3″:4′,5′]thieno[2′,3′:4,5]pyrrolo[3,2-g]thieno[2′,3′:4,5]thieno[3,2-b]indole-2,10-diyl)bis(methanylylidene))bis(5,6-difluoro-3-oxo-2,3-dihydro-1H-indene-2,1-diylidene))dimalononitrile (Y6). PEDOT:PSS acts as hole transport layer and facilitates the wettability of the BP ink. Meanwhile, Y6 is an electron acceptor material commonly used in high-performance organic optoelectronics[30]. Importantly, automatic film applicators can be used to effectively deposit PEDOT:PSS and Y6 solutions[31,32], ensuring the scalability of the fabrication. Indium tin oxide (ITO) was used as the top electrode, whereas Au served as the bottom electrode, resulting in a vertical device structure of Au/PEDOT:PSS/BP/Y6/ITO (see Supplementary Fig. 1 for band structure alignment). The imager (illustrated in Fig. 1b) was fabricated by arraying the Au bottom electrodes which define the active area of photodetection by their overlapping area with the top ITO electrodes. Ti/Au pads and readout lines were deposited via evaporation, and the readout lines were coated with $SiO_2$ to prevent pixel crosstalk.

A wafer-scale coating of the BP ink film on a flexible plastic substrate was demonstrated. Figure 1c shows the backside of an imager array fabricated on a transparent and flexible substrate, where the BP ink film is visible in black (see Supplementary Fig. 2 for the frontside). The inset figure shows the optical micrograph of a single pixel after the Y6 and stripe-patterned ITO deposition. For the deposition, a scalable process for forming the BP film using an automatic film applicator was demonstrated (see "Methods" for deposition recipes of the BP ink). Moreover, the Raman spectrum of the BP ink displayed characteristic peaks at approximately 364, 440, and 467 $cm^{-1}$, corresponding to known $A_g^1$, $B_g^2$, and $A_g^2$ phonon modes of bulk BP, respectively[33] (see Supplementary Fig. 3). The integrated intensity ratio of $A_g^1$ to $A_g^2$ modes was approximately 0.52. This ratio, which exceeds 0.2, indicates minimal oxidation of the BP film during ink fabrication and film formation[34].

Figure 1d shows the top-view micrograph of the BP film coated on polyimide (PI)/Au/PEDOT:PS captured using scanning electron microscopy (SEM), highlighting tile-like packing of the BP nanosheets. Figure 1e shows a false-colored cross-sectional SEM micrograph of BP film deposited on PI/Ti/Au/PEDOT:PSS. By tuning the concentration and volume of the applied ink, the film thickness was controlled to achieve the thickness of approximately 1.0 $\mu m$. This thickness was found to reproducibly yield films without pinholes across the entire imager. Figure 1f shows the thickness mapping of an 8 mm × 8 mm BP film, where the average thickness was 1.1 $\mu m$.

## MWIR photodetection

The *I-V* characteristics of the photodetector showed distinct differences between dark and illuminated conditions, exhibiting diode-like behavior, with an open-circuit photovoltage and a short-circuit photocurrent ($I_{ph}$) (Fig. 2a). MWIR detection up to $\lambda = 4$ $\mu m$ was confirmed using a custom-built Fourier transform infrared (FTIR) detector characterization platform, corresponding to a bandgap energy larger than 0.31 eV. Figure 2b presents the responsivity versus photon energy for the unbiased device. The measured FTIR spectrum was first normalized to the response of a deuterated triglycine sulfate (DTGS) detector and subsequently scaled to match the responsivity obtained under $\lambda = 1.55$ $\mu m$ irradiation using a laser diode. The laser diode delivered an incident power density of 1.6 $W \cdot cm^{-2}$, comparable to that of the FTIR source at the same wavelength, as indicated by a similar measured photocurrent. This correspondence enabled accurate scaling of the FTIR spectrum.

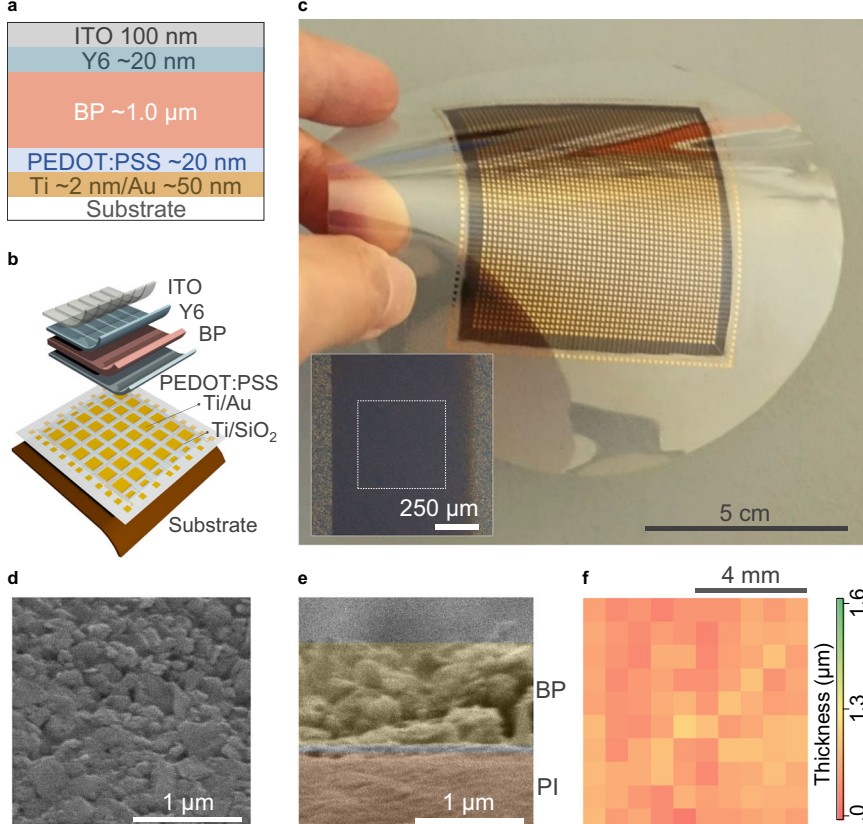

**Fig. 1 | Black phosphorus (BP) ink film for mid-wave infrared (MWIR) imager.** Schematic illustration of the cross-sectional structure of (**a**) a single-pixel photo-detector and (**b**) the imager. ITO, indium tin oxide; Y6, a non-fullerene acceptor; PEDOT:PSS, poly(3,4-ethylenedioxythiophene):poly(styrenesulfonate). **c** Optical image of the backside of an imager array fabricated on a transparent, flexible film, with photodetector pixels incorporating BP films. The bottom Au electrodes are routed to external Au pads positioned around the perimeter of the array. The black film is the BP film, functioning as the photoactive material of the photodetectors, while other functional layers are not optically distinguishable. (Inset) Optical micrograph of single pixel. The light gray area represents the PI/Ti/Au/PEDOT:PSS/BP/Y6 layers, while the dark gray area includes an additional ITO film layer. The white dashed line outlines the pixel. **d** In-plane optical micrograph of the BP film deposited on a PI/Ti/Au/PEDOT:PSS stack. **e** Cross-sectional optical micrograph of a photodetector pixel deposited on a PI/Ti/Au/PEDOT:PSS substrate. **f** Thickness mapping of an 8 mm × 8 mm BP film deposited on a quartz/Ti/Au/PEDOT:PSS substrate.

The device exhibited a responsivity of 45 μA·W$^{-1}$ at $\lambda = 1.55$ μm and 20 μA·W$^{-1}$ at $\lambda = 3.0$ μm. The FTIR-derived spectrum demonstrated the MWIR response of the device, with measurable responsivity extending up to the BP bandgap (~0.33 eV). Instead of a sharp cutoff near the band edge, the spectrum showed a gradual decline. To further validate MWIR sensitivity, the device was irradiated with a $\lambda = 3.3$ μm LED at a power density of 33 mW·cm$^{-2}$, yielding a measured responsivity of 4.2 μA·W$^{-1}$ (Supplementary Fig. 4). Based on the spectral characteristics and power dependence (given below), the expected responsivity at $\lambda = 3.3$ μm and power level was estimated to be 4.8 μA·W$^{-1}$, corresponding to a 14% deviation of the measured value. Following confirmation of the MWIR detection, subsequent characterizations were performed using the 1.55-μm laser.

Figure 2c shows that reverse biasing the device amplified the photoresponse, with the responsivity at $\lambda = 1.55$ μm increasing from 45 μA·W$^{-1}$ at 0 V to 2.0 mA·W$^{-1}$ at −2.5 V, before saturating at larger reverse biases. This behavior is characteristic of a conventional pn-junction photodiode, Supplementary Fig. 1 presents the band diagram of the photodiode between the two electrodes with the junction situated at the BP/Y6 interface. Reverse bias enhances the electric field across this interface, facilitating more efficient separation of photogenerated carriers, consistent with observations in similar systems employing planar junctions of solution-processed BP and organic electron acceptors[35].

Additionally, the photocurrent of the unbiased device exhibited a linear dependence on incident power over the range of 17 μW to 6.0 mW (Fig. 2d). The responsivity can be further improved by reducing defective surfaces and edges when oxidized layers of PO$_x$ formed on the BP nanosheets during the ink or device fabrication process, which potentially act as recombination centers for charge carriers[36,37]. Moreover, passivating defects introduced by remnants of organic solvents or binder agents of the ink offers a way to suppress noise current[38,39]. Detailed analysis of the noise characteristics and specific detectivity ($D^*$) of the BP photodetector is provided in Supplementary Note 3 and Supplementary Fig. 5.

The rise and fall time of the device under $\lambda = 1.55$-μm irradiation at 0 V was 4.1 μs and 2.5 μs, respectively (Fig. 2e). These values are comparable to the fastest reported flake-based BP-based photoconductors and photodiodes[19,40,41] (Supplementary Table 3). Irradiation duration exceeding the transient response times of the BP photodetector confirmed the absence of a long-tail slow response (Supplementary Fig. 6). Additionally, the reproducibility of the photoresponse was verified through consistent behavior across multiple switching cycles (Supplementary Fig. 7).

Figure 2f shows the frequency response of the device at 0 V, showing the 3 dB frequency at approximately 44 kHz. The calculated 3 dB frequency using the standard resistor-capacitor (RC)-limited model, based on the rise time, is approximately 86 kHz, which is higher than the measured value. This discrepancy suggests that additional factors influenced the response time. Potential factors include surface defects at the BP/Y6 interface or within Y6 layers[42], as well as inefficient charge carrier extraction at the electrode contacts[43], both of which

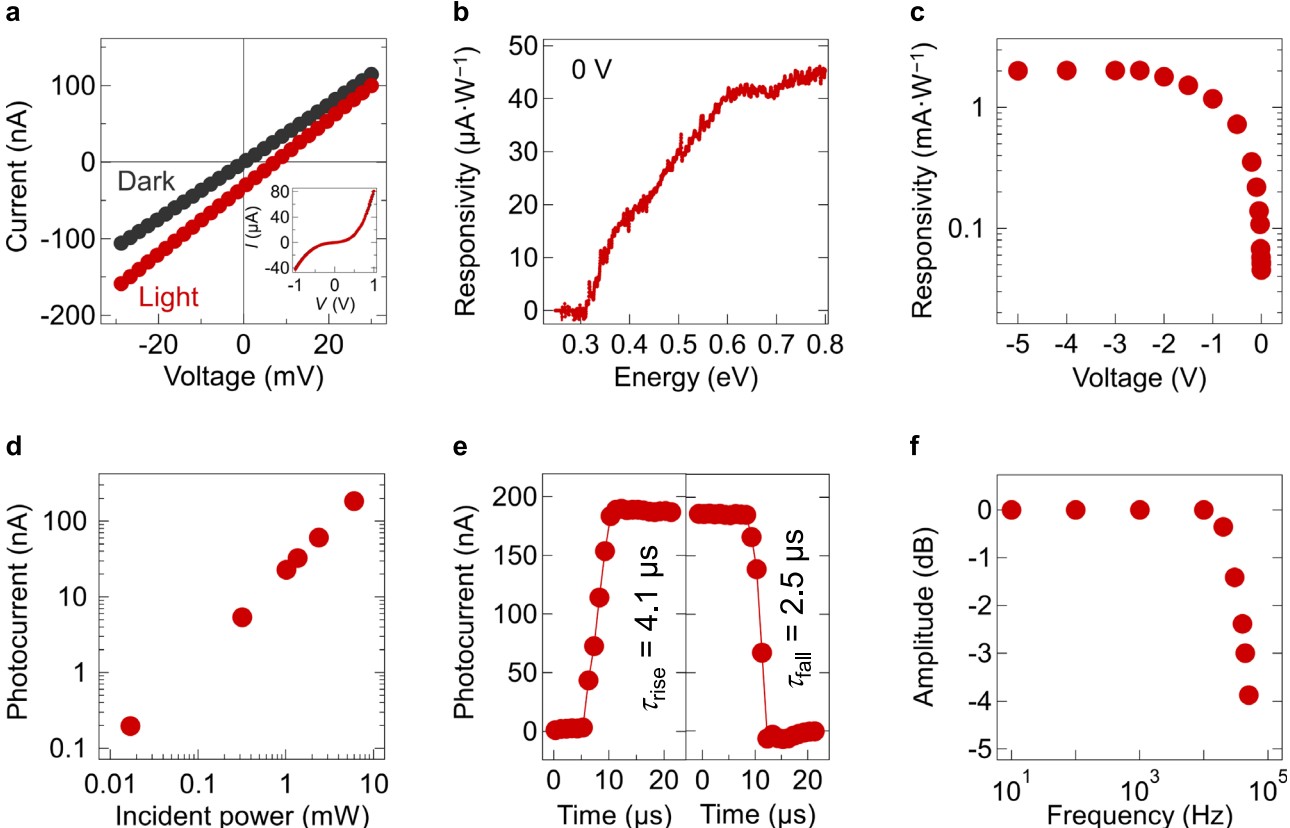

**Fig. 2 | Characteristics of a single-pixel BP photodetector. a** Current vs. voltage ($I$–$V$) characteristics in the dark (black) and under light irradiation (red). The light source is a blackbody radiation (1000 K, 9 mW·cm$^{-2}$). The inset shows the $I$–$V$ characteristics measured over a broader bias range. **b** Optical spectra of the photodetector under zero bias. The photocurrent at each wavelength is calibrated using the photocurrent measured under infrared irradiation at $\lambda = 1.55$ μm (1.6 W·cm$^{-2}$). All subsequent characterizations used the 1.55-μm laser diode as the light source. **c** Responsivity as a function of applied voltage. **d** Power dependence of the photodetector. **e** Rise and fall times of the photodetector at 0 V, measured using the 90–10% method. **f** Frequency response of the photodetector at 0 V, showing a cutoff frequency at 44 kHz.

were reported in analogous systems to suppress high-frequency response[44].

## MWIR imaging

Figure 3a shows an optical photograph of a BP imager array, comprising 6 × 6 pixels, mounted on a chip carrier and encapsulated by adhering a quartz window with an ultraviolet-curable resin in a nitrogen-purged environment. The imager is schematically illustrated in Fig. 3b, with the array design shown in Supplementary Fig. 8. Figure 3c presents an illustration of the imaging setup configuration. Imaging was performed with a shadow mask placed directly on top of the imager chip for selective imaging with a $\lambda = 1.55$ μm laser, as detailed in the "Methods" section.

The signal from each photodetector was measured at 0 V and calibrated to output the measured optical power during imaging. The 36 unbiased pixels exhibited a responsivity of $23 \pm 7$ μA·W$^{-1}$ under $\lambda = 1.55$ μm irradiation. Supplementary Fig. 9 presents the spatial mapping of responsivity across the imager pixels and the corresponding frequency distribution.

For imaging, three shadow masks with patterns of the letters "C," "A," and "L" were used, and the imaging results showed a recognizable representation of the letters (Fig. 3d). Supplementary Fig. 10 illustrates the positions of the pixels under the shadow masks and compares the expected and measured imaging results. The error percentages calculated from the differences between the expected results (determined by the overlap area of the mask and the imager) and the measured signals were 0.8%, 1.1%, and 0.4% for the letters "C," "A," and "L," respectively, demonstrating reliable imaging performance. The imager was also tested

under MWIR irradiation using a $\lambda = 3.3$ μm LED. Supplementary Fig. 11 presents the corresponding imaging pattern and results, which showed a clear reconstruction with an error percentage of 5.3%.

The long-term stability of the photodetector was evaluated under both air storage and continuous irradiation conditions. Figure 3e shows the responsivity comparison of encapsulated and non-encapsulated devices. The responsivity of the non-encapsulated device decreased to 48% after 312 h (13 days), whereas the encapsulated device maintained its full responsivity even after 3600 h (150 days) of storage. The spectral difference between the initial measurement and that after 336 h (14 days) was minimal, confirming the spectral stability of the encapsulated device (Fig. 3f). Minor spectral variations were attributed to slight differences in sample positioning during measurement.

Additionally, the operational stability of the device under constant irradiation was evaluated using a 400 °C blackbody source at a power density of 0.19 W·cm$^{-2}$, as shown in Supplementary Fig. 12. Under this condition, the encapsulated device maintained its responsivity over 120 h (5 days), while the responsivity of the non-encapsulated device degraded to 40% of its initial value after 25 h (~1 day) of continuous exposure.

## Mechanical flexibility of the photodetector

Two thicknesses of the PI substrate were used in the tests: 25 μm and 125 μm. The 25-μm PI was used for tests with varying bending radii, as it allows a broader range of bending with minimal strain. In contrast, the 125 μm PI was used for cyclic bending tests at a 1-cm radius, as its thickness facilitates easier attachment to the mechanical flexing setup. Figure 4a shows this setup, where one edge of the film is attached to a

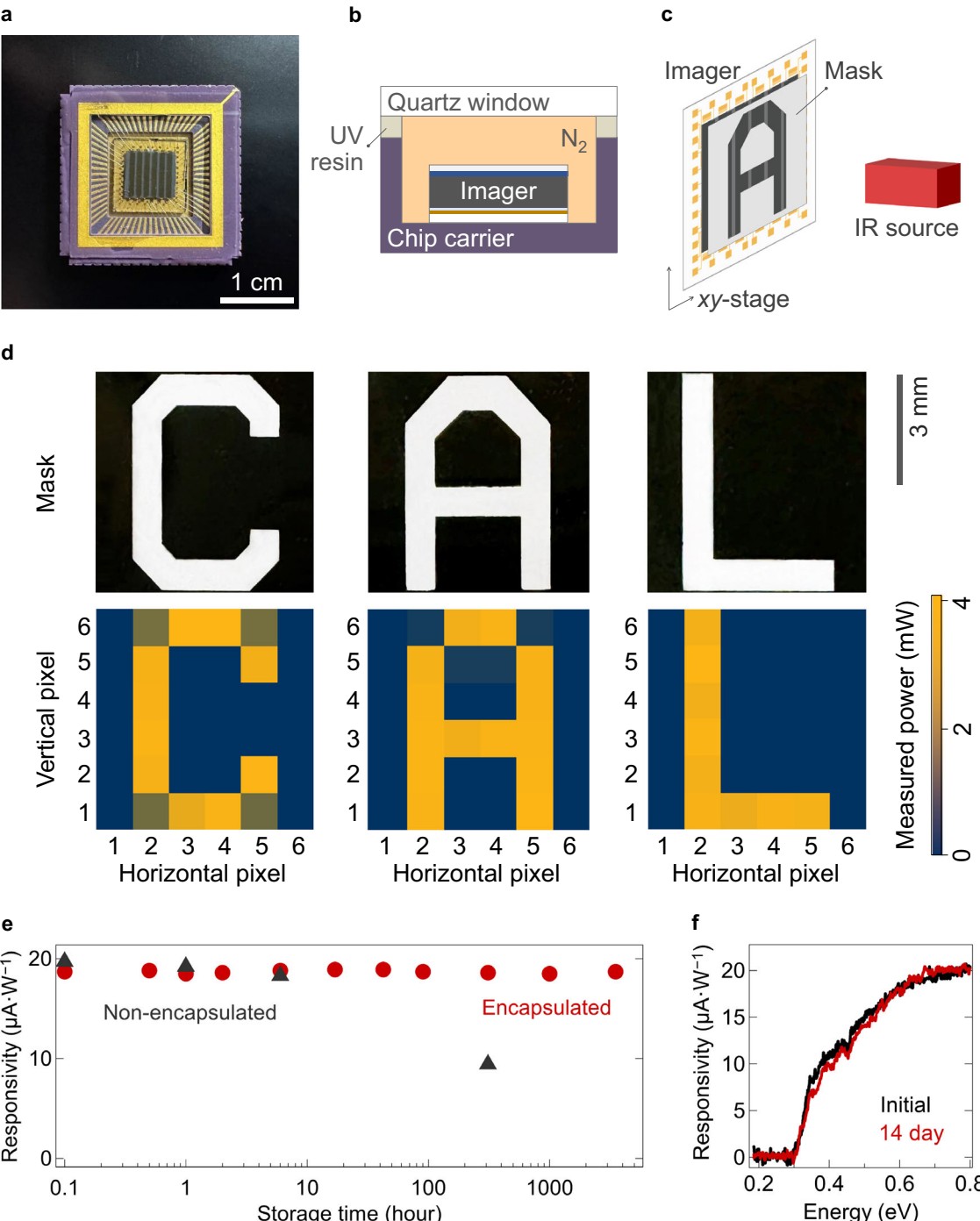

**Fig. 3 | MWIR imaging using an array of BP photodetectors. a** Optical photograph of the encapsulated imager mounted on a chip carrier. The BP film appears gray, while the top ITO electrodes, deposited across the columns, appear dark gray. **b** Schematic illustration of the encapsulated imager. The device is packaged in a nitrogen environment (colored orange for emphasis), using a quartz window and ultraviolet-curable resin for sealing. The wire lines connecting the imager to the chip carrier are omitted for simplicity. **c** Schematic illustration of the imaging setup. A shadow mask is used to selectively irradiate the imager with a $\lambda = 1.55$-μm laser diode. The imager is moved precisely using an *xy*-stage for pixel-by-pixel imaging. **d** (Top) Optical photograph of the masks, made of quartz/Ti/Au and patterned with the letters "C," "A," and "L." (Bottom) Imaging results of letter patterns, presented as measured optical power per pixel. **e** Responsivity during the air-storage stability test of the encapsulated and non-encapsulated photodetector. **f** Optical spectra of the encapsulated photodetector, before and after the air-storage stability test.

---

fixed support and the other to a movable slab to adjust the bending radius. For devices fabricated on a 25 μm PI film, the flexibility was tested by directly flexing the devices while conformally attaching them to a bent surface with different radii. Figure 4b presents the bending test results, with the photodetector fabricated on a 25-μm PI film exhibiting stable responsivity, maintaining its original value even after bending at 2.5-mm radius, highlighting the excellent mechanical

flexibility of the device. The same device fabricated on 125 μm PI maintained its responsivity after bending at a 1-cm radius, but the responsivity gradually decreased to 41% of its original value after bending at a 2.5-mm radius, owing to the higher strain induced by flexing the thicker substrate (Supplementary Fig. 13). However, the flexibility test on PI (125 μm)/PEDOT:PSS/BP film showed that the BP film maintained 97% of its conductivity after 2.5-mm bending

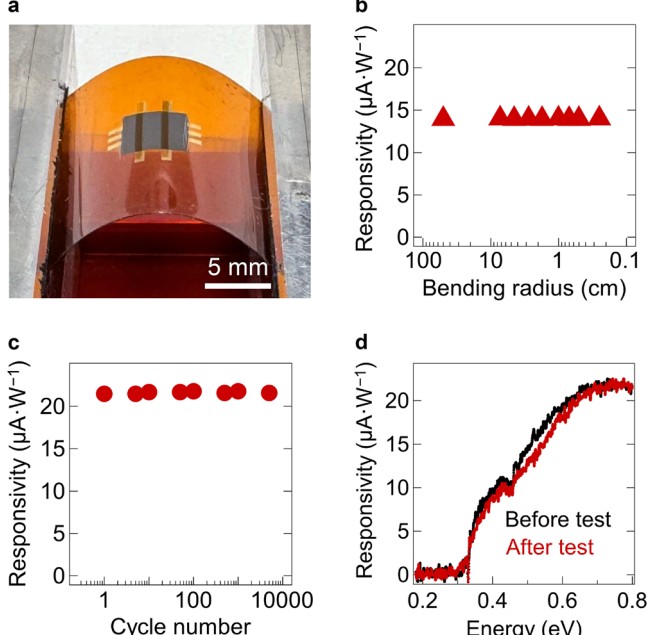

**a** ... **5 mm**

**b** Responsivity ($\mu A \cdot W^{-1}$) vs Bending radius (cm)

**c** Responsivity ($\mu A \cdot W^{-1}$) vs Cycle number

**d** Responsivity ($\mu A \cdot W^{-1}$) vs Energy (eV) — Before test / After test

**Fig. 4 | Mechanical flexibility test of a BP photodetector. a** Optical photograph of the setup for the mechanical flexibility test. **b** Responsivity values measured during the bending test of the BP photodetector fabricated on PI film (thickness: 25 μm). **c** Responsivity values measured during the cyclic bending test of the BP photodetector on PI film (thickness: 125 μm) with a 1-cm bending radius. **d** Optical spectra of the photodetector before and after 5000 bending cycles.

(Supplementary Note 2 and Supplementary Fig. 14), implying that the BP film exhibits robust mechanical stability and does not intrinsically restrict the mechanical flexibility of the device. On the other hand, the ITO electrode used in the device has a limited mechanical flexibility[45]. Hence, replacing it with more mechanically flexible electrode materials, such as carbon nanotubes[46], provides a solution to further enhance the flexibility of the device when a thicker substrate is necessary.

In another test, the 125 μm PI substrate device was subjected to a cyclic bending with a 1-cm bending radius. The results demonstrated that the device maintained its stability even after 5000 bending cycles (Fig. 4c). Additionally, the optical spectrum of the device before and after 5000 bending cycles showed no significant change in shape (Fig. 4d), with the minimal variations attributed to slight differences in the irradiation location on the device. Therefore, the device exhibited stable mechanical flexibility up to bending radii of 2.5 mm on thin substrates and after thousands of bending cycles on thicker substrates.

The BP ink film and the device structure developed in this study successfully enabled flexible MWIR imagers, filling the technological gap in mechanical flexibility and operation at 0 V, which were not addressed by the previously reported BP ink photodetectors[24,29,47,48] (detailed further in Supplementary Table 2). The flexibility of the imagers, combined with their use of non-toxic compounds and demonstrated scalability, distinguishes them from other material systems and commercial products (detailed further in Supplementary Table 3). Specifically, colloidal quantum dots systems using PbS/HgTe have also enabled high-performance MWIR photodetection at room temperature[10,49], and their fabrication processes are theoretically compatible with the process requirements of flexible electronics. However, Hg-based compounds pose toxic risk to both humans and the environment, with regulatory measures being implemented to phase out or reduce the use of Hg-based materials[50]. In this context, the BP ink-based system and device architecture developed in this study offer a viable solution for flexible MWIR photodetectors.

## Imaging with flexible imagers

To demonstrate the application of the proposed system, a flexible MWIR imager consisting of a 1 × 8 photodetector linear array was fabricated using a BP ink film on a 25 μm PI substrate. Figure 5a shows an optical photograph of the imager attached to a curvilinear surface with a radius of 7.5 mm. The pixels were positioned equidistantly at 45° intervals. For simplicity, ITO was deposited across the entire vertical stack, with its contact extended beyond the BP film to overlap with an Au pad. The Au pads of the devices and the ITO contacts were connected to extended Au electrodes on a separate PI film using anisotropic conductive film (ACF) tape. Figure 5b schematically illustrates this design from both front and back perspectives.

Pixels labeled as Pixel 1 through Pixel 8 were irradiated using a broad-spectrum light source (2796 K, 0.23 W·cm$^{-2}$) with a Gaussian intensity profile to stimulate simultaneous photoresponse from multiple pixels. The light source was positioned parallel to the pixel array, ensuring that at least three adjacent pixels were irradiated at any given time. It was then rotated axially around the imager, with the angular position ($\theta$) measured relative to the center of Pixel 1. The imaging setup is schematically illustrated in Fig. 5c.

Figure 5d shows the photocurrent as a function of the light source angle across a full 360° rotation, measured at 15° intervals. For instance, at $\theta = 0°$, photocurrent was detected from Pixel 1, Pixel 2, and Pixel 8, whereas the remaining pixels were outside the irradiation range and, consequently, showed no response. Pixel 1 exhibited its maximum photocurrent under orthogonal irradiation at $\theta = 0°$ and ceased responding when the light source angle was between 90° and 270° (i.e., when $\theta \equiv -90°$ (mod 360°)). All pixels demonstrated a similar photoresponse profile as a function of light source angle. Supplementary Fig. 15 presents spatial photocurrent mapping at each angular position of the light source.

From the imaging results, the light source angle ($\theta$) was reconstructed by fitting the detector readout $I_{ph,i}(\theta)$ of the $i$-th pixel ($i = 1, 2, ..., 8$) with Gaussian beam profiles, where $\theta$ represents the angular position of the pixels. Specifically, center angle $\theta_0$, peak width $\sigma$, and amplitude $A$ of the Gaussian beams were used as fitting variables in a least-square regression model:

$$\min_{\{A,\,\theta_0,\,\sigma\}} \sum_{i=1}^{8} \left( I_{ph}(\theta_i) - A \exp\left( -\frac{(\theta_i - \theta_0)^2}{2\sigma^2} \right) \right)^2.$$

The optimized $\theta_0$ for the eight detectors within a single frame corresponds to the reconstructed light source angle, with a comparison between the ground truth and the predicted value as shown in Fig. 5e. The root mean square (RMS) error of the angle prediction was 2.9°, demonstrating reliable full-azimuthal prediction of the light source angle. For an unknown light source, a more densely packed imager array can improve the accuracy of the initial fitting function for the prediction model, enabling similar prediction to be performed.

Additionally, multirow−multicolumn imaging was demonstrated using a 6 × 3-pixel flexible imager conformally attached to a curvilinear surface with a 7.5 mm radius. The six rows spanned a length of 1.8 cm, and the three columns were spaced at 45° intervals. Figure 6a shows an optical photograph of the two-dimensional scanner on the curved surface, while Fig. 6b illustrates the device structure and its extended electrode film. Figure 6c, d present schematics of the full-field irradiation setup using the same broad-spectrum light source, viewed from the side and front, respectively. In this configuration, the center row was orthogonally irradiated, and all pixels were exposed. All 18 pixels generated photocurrent, with the edge-column pixels detecting 36 ± 3% of the maximum optical power measured under orthogonal irradiation (Fig. 6e). Figure 6f, g illustrate the selective irradiation of the scanner, where Rows 3 and 5 received less exposure than Row 4. The corresponding imaging result is shown in Fig. 6h, where pixels in

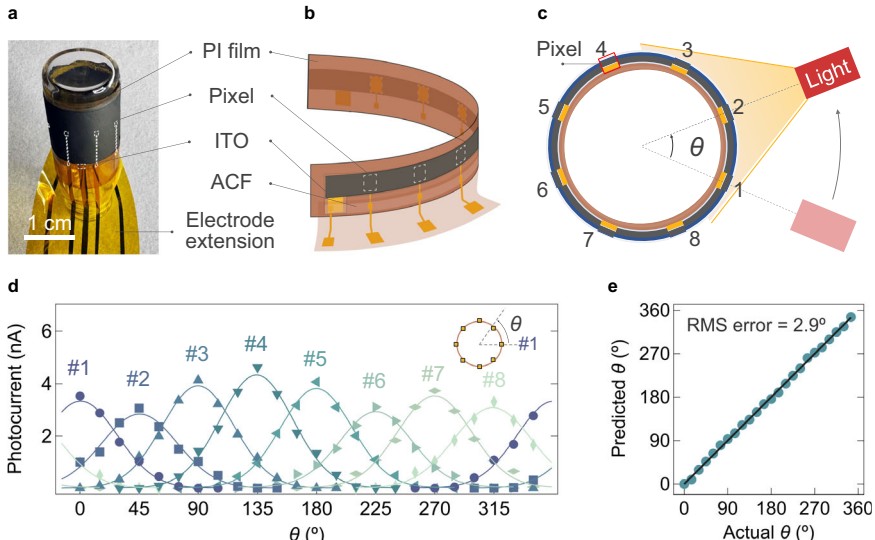

**Fig. 5 | Panoramic wide-angle imaging using flexible BP imager. a** Optical photograph of a flexible imager with equidistantly positioned 1 × 8 pixels fabricated on a 25 µm-thick polyimide (PI) film, attached to a curvilinear surface (radius: 7.5 mm). The white dashed lines outline the imager pixels. **b** Schematic illustration of the flexible imager, showing the device structure from both sides. The white dashed line delineates the imager pixels. The PI film with extended electrode pads is attached to the imager using anisotropic conductive film (ACF) tape (shown for the front half of the imager for clarity). **c** Schematic illustration of the imaging setup, viewed from above. The light source is moved through axial rotation around the imager, with the angular position (θ) measured relative to the center of Pixel 1. The light source is positioned 1 cm from the imager. **d** Photocurrent of the eight pixels as a function of the angular position of the light source, measured in 15° intervals. Solid lines are visual guides to indicate data trends. The light source is a tungsten halogen lamp (2796 K, 0.23 W·cm⁻²). (Inset) Schematic illustration of the light source angle θ, defined with respect to the normal of Pixel 1. **e** Predicted versus actual light source angle (θ), reconstructed by fitting detector signals to angular profiles using a least-squares regression model. The root mean square (RMS) error of 2.9° demonstrates reliable full-azimuthal detection of MWIR light. The solid line represents the ideal case, where predicted angle equals to actual angle.

Rows 3 and 5 exhibited 51% and 61% of the optical power measured in Row 4, respectively, demonstrating accurate reconstruction of the selective irradiation pattern.

These demonstrations highlight the advantages of the ink-based BP film, which not only enables an expanded field of view for the imager through the mechanical flexibility of the entire structure, but also facilitates a single-substrate, monolithic fabrication of a multipixel imager owing to the scalability of the solution-based process.

## Discussion

In this study, we demonstrated a self-powered, flexible MWIR imager that operates at room temperature by using BP ink films. Specifically, the photodetector exhibited responsivities ranging from 20 µA·W⁻¹ at $\lambda = 3.0$ µm to 0.3 µA·W⁻¹ at $\lambda = 4.0$ µm. This achievement was realized by adjusting the BP ink dimensions to achieve an effective MWIR detection, and by designing the interlayers to allow uniform coating over large area. Additionally, the device fabricated on a thin PI film exhibited excellent mechanical flexibility, as demonstrated by stable performance during bending tests with radii as small as 2.5 mm and after thousands of bending cycles with a 1-cm radius. A flexible imager composed of a 1 × 8 array of photodetectors enabled 360° panoramic imaging, expanding the field of view of the imager through its mechanical flexibility. This design also maintained the advantages of a unified, monolithic structure, allowing for efficient fabrication on a single substrate. Notably, the ink-based BP film allowed for the fabrication of a large-scale MWIR imager on various substrates, including plastic films, reducing both thermal budget and the fabrication cost. This study provides a scalable, substrate-compatible fabrication process for flexible MWIR imagers.

## Methods
### Ink fabrication
All procedures were conducted inside a nitrogen-filled glovebox. Black phosphorus (BP) crystals (Smart Elements) were ground into a fine powder using a mortar and pestle. The powder was then dispersed in anhydrous 1-methylpyrrolidin-2-one (NMP, Sigma Aldrich) at a concentration of 0.3 g/mL. The dispersion was ultrasonicated using a 150-W Cole Parmer Ultrasonic Processor for 2 h at 60% amplitude with a 50% duty cycle (1 s on, 1 s off). Following sonication, the dispersion was centrifuged at 250g for 10 min. The supernatant was collected and further centrifuged at 4014g for 30 min. After carefully removing the NMP, the BP sediment was mixed with oleylamine (Sigma Aldrich) at a ratio of 0.1 µL per 25 mg of BP, then redispersed in toluene (Sigma Aldrich) at a concentration of 400 µL per 25 mg of BP. The mixture was bath-sonicated at low power (Crest Ultrasonics 275DA, power level 1) for 5 min and centrifuged again at 4014g for 30 min. The resulting sediment was collected and redispersed in toluene at a ratio of 800 µL per 10 mg of BP. Complete dispersion was ensured by low-power bath sonication, taking care to prevent any residual BP from adhering to the microtube walls.

### BP film characterizations
Raman spectroscopy was performed using a Raman microscope system (Labram HR Evolution, Horiba) with an excitation laser ($\lambda = 532$ nm) and a spot size of approximately 5 um. The thickness of the BP film was determined by measuring the step height of the BP film on a quartz/PEDOT:PSS substrate using a stylus profilometer. Top-view and cross-sectional micrographs of the BP films were taken using a scanning electron microscope (FEI Quanta 600 F).

### Device fabrication
The rigid devices were fabricated on cut 1 cm × 1 cm quartz substrates. For the flexibility test, the devices were fabricated on PI films with thicknesses of 25 µm or 125 µm. The substrates were cleaned using deionized water, acetone (Sigma Aldrich), and 2-propanol (Sigma Aldrich). The bottom electrodes were made by depositing Ti (2 nm) and Au (50 nm) sequentially using standard evaporation. The device area used for basic characterization was 500 µm × 500 µm, defined by

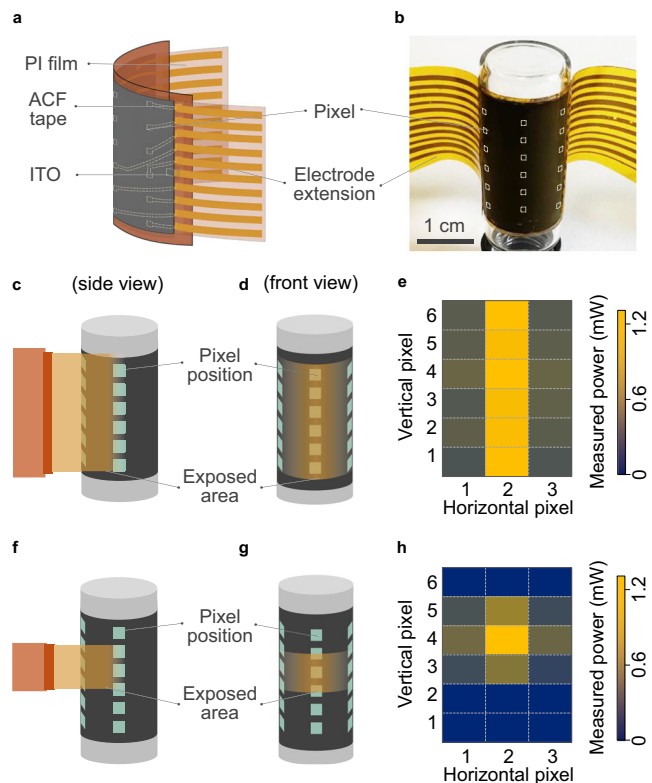

**Fig. 6 | Two-dimensional scanner using a flexible BP imager. a** Schematic of the flexible imager showing its structure from both sides. The PI film with extended electrode pads is connected to the imager using ACF tape. **b** Optical photograph of a 6 × 3-pixel flexible imager fabricated on a 25 μm-thick PI film and conformally attached to a curvilinear surface (radius: 7.5 mm). The white dashed line outlines the imager area. **c**, **d** Schematic illustration of the device under full-field irradiation by a tungsten halogen lamp (2796 K, 0.23 W·cm⁻²), viewed from the side (**c**) and front (**d**). Cyan squares indicate the pixel positions. The light source is positioned 1 cm from the device. **e** Imaging result under full-field irradiation, presented as measured optical power per pixel. **f**, **g** Schematic of the device under selective irradiation, viewed from the side (**f**) and front (**g**). **h** Imaging result under selective irradiation, presented as measured optical power per pixel.

the intersectional area between the bottom Au electrode and the top ITO electrode. The samples were cleaned and treated using O₂ plasma at 300 W for 10 s. PEDOT:PSS solution (Ossila AI 4083) was prepared by diluting to 15% by volume in anhydrous 2-propanol (Sigma Aldrich). Films were formed by filtering (0.45-μm filter) the PEDOT:PSS solution and dropping to cover the whole sample area, followed by spincoating at 2000 rpm for 30 s, and annealing at 150 ℃ for 10 min. All annealing processes were done inside a nitrogen-purged glovebox and were followed by a gradual cooling to 60 ℃ before the samples were removed. The BP film was formed using an automatic film applicator (Proceq ZUA 2000, Supplementary Fig. 16), where the blade coated the substrate with ink (2.5 μL for a 1 cm × 1 cm substrate) at a speed of 1 cm·s⁻¹ for several times. The film was immediately annealed at 180 ℃ for 10 min. Alternatively, spin-coating the BP ink (22.3 μL for a 1 cm × 1 cm substrate) at 60 rpm for 5 min, followed by an annealing at 180 ℃ for 1 h, also produced a BP film for MWIR photodetectors with similar performance. Y6 solution (1-Material) was prepared by dissolving it in anhydrous chloroform (Sigma Aldrich) with a concentration of 9 mg/ 1 mL, followed by stirring at 300 rpm while being heated at 50 ℃ for 4 h. Y6 solution (50 μL/1 cm²) was dynamically spincoated at 3500 rpm for 45 s, followed by annealing inside a glovebox at 110 ℃ for 10 min. The PEDOT:PSS/BP/Y6 film was partially patterned to uncover the Au readout pads by using acetone. Finally, the top electrodes were made by sputtering ITO (AJA International) at 30 W.

Basic imaging was performed using the imager fabricated on quartz substrate, composed of 6 × 6 photodetectors with a single-pixel active area of 750 μm × 750 μm. The encapsulation of the imager device was performed inside a glovebox using a z-cut, double-sided polished quartz sheet (500 μm), which was attached to the chip carrier with a UV-sensitive resin (Norland Optical Adhesive 85). The flexible imagers were fabricated on 25-μm PI films with a single-pixel active area of 750 μm × 750 μm. During fabrication, the PI film was temporarily attached to a quartz support. The area except the Au pads was covered with Ti (2 nm)/SiO₂ (50 nm) to prevent cross-talking between pixels. All functional layers were deposited in the same manner as for the rigid devices. To minimize potential mechanical stress on the device caused by probing during measurements, the flexible imager film was separated from the probing film. This separation was achieved by extending the Au pads of the imagers to additional Au pads on another 25 μm PI film, which was attached to the imager using ACF tape.

## Photodetector characterization

Devices were first wire-bonded onto chip carriers and installed into a chamber (Lakeshore Janis) maintained at ambient pressure. Current–voltage (I–V) characteristics were recorded using a semiconductor parameter analyzer (Agilent B1150A). To evaluate photocurrent behavior (Fig. 2a), the samples were illuminated with broadband infrared radiation from the internal Globar source of an FTIR system (1000 K, 9 mW·cm⁻² at the output port). For wavelength-specific response measurements, a 1.55 μm laser diode (Thorlabs L1550P5DFB, 1.6 W·cm⁻²) was modulated at 10 Hz using a laser controller (Thorlabs ITC4002QCL) and a function generator (Agilent 33120 A). The resulting photocurrent was passed through a low-noise current amplifier (SRS570) and processed by a lock-in amplifier (SR865) synchronized to the modulation frequency. All measurements were performed at 0 V bias unless otherwise specified. For the data in Fig. 2c, a voltage bias was applied using the internal source of the amplifier. Spectral response was obtained under zero bias by positioning the device at the FTIR's external port (Thermo Fisher Nicolet iS50), with the output current routed through the SRS570 and into the FTIR's external detector input. The FTIR spectrum result was calibrated using a built-in DTGS detector was used to calibrate the relative spectral intensity of the source, as also detailed in ref. 19. Additionally, an LED with λ = 3.3 μm (Hamamatsu L15893-0330ML, 33 mW·cm⁻²) was used to confirm the MWIR response.

To assess linearity, the incident power density was tuned by adjusting the drive current of a 1.55 μm laser source. Detector response speed was characterized using a 10-kHz modulated laser beam, and rise/fall times were extracted based on the 90–10% criterion. The resulting photocurrent signal was amplified using an SRS570 current amplifier and monitored with a Tektronix TDS3000B oscilloscope, triggered by the modulation signal from the function generator. For frequency response analysis, the 3 dB cutoff was determined using the current amplifier operated in high-bandwidth mode (gain > 20 μA·V⁻¹, corresponding to ~1 MHz bandwidth), followed by lock-in detection. Noise spectral density was measured using the same lock-in amplifier setup under dark conditions.

## Imaging

Shadow masks made of quartz (500 μm)/Ti (5 nm)/Au (100 nm) with letter patterns were used to selectively irradiate the 6 × 6 imager by placing them directly on top of the imager. A coordinate-controlled translating sample stage precisely moved so that the λ = 1.55 μm laser (1.6 W·cm⁻²) irradiated the center of each pixel. The laser was sequentially scanned across all 36 photodetectors, and the corresponding photoresponses were recorded. The imaging results account for pixel variation (Supplementary Fig. 9) by calibrating each pixel using the photocurrent versus incident optical power response of a reference pixel with average responsivity.

For imaging with flexible imagers, the light source was a tungsten halogen lamp operated at 2796 K (Thorlabs SLS2021, 0.23 W·cm$^{-2}$), providing broad irradiation to simultaneously illuminate multiple pixels of the imager. The light was modulated at 1 kHz using a chopper. For irradiation at steep angles, the light source angle $\theta$ was defined as the angle between the normal vector of the imager midpoint (Pixel 1) and the vector pointing toward the light source.

## Stability & mechanical flexibility tests

In the air-storage stability tests, the encapsulated device was kept in the dark under air-conditioned laboratory environment and was exposed to external light irradiation only during measurements using a $\lambda = 1.55\,\mu m$ laser. For the operational stability tests, a 400 °C black-body source (0.19 W·cm$^{-2}$) modulated by a 300 Hz chopper was used. In the mechanical flexibility tests of the photodetectors, the two edges of the 125-μm thick PI substrate were securely attached, one to a fixed support and another to a movable slab (Fig. 4a) which allows precise control over the flexing radius of the film. The 25 μm PI film was flexed by attaching the device directly on a curved surface with varying radii. The electrodes were extended using ACF tape and a separate film with contact pads for measurements.

## Data availability

The Source Data underlying the figures of this study are available with the paper. All raw data generated during the current study are available from the corresponding authors upon request. Source data are provided with this paper.

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

## Acknowledgements

Material characterizations and optical measurements were supported by the U.S. Department of Energy, Office of Science, Office of Basic Energy Sciences, Materials Sciences and Engineering Division under contract No. DE-AC02-05Ch11231 (Electronic Materials Program). T.J.W. acknowledges fellowship support from JSPS KAKENHI Grant Number JP23KJ0760 and Mizuho Foundation for the Promotion of the Sciences. N.H. acknowledges support from JST PRESTO (JPMJPR23H7), Japan for part of the device fabrication. The automatic film applicator experiments were conducted in the Printed Electronics Lab of the Arias Research Group, UC Berkeley, with help from S. Koh and P. Sahinidis.

## Author contributions

T.J.W., N.H., T.Y., T.S., and A.J. conceived the project and designed the experiments. T.J.W., N.H., S.T., H.M.K., and D.Z. formulated the BP ink. T.J.W., N.H., and Shifan W. designed the imager. T.J.W. and S.T. fabricated the samples. T.J.W., N.H., and S.T. measured the devices. T.J.W., N.H., Shifan W., S.T., H.M.K., Shu W., and D.Z. built the optoelectronic characterization setup. H.M.K. performed the Raman spectroscopy. Shu W. performed the SEM analysis. T.J.W., N.H., and H.M.K. took optical photographs. D.Z. assisted with the flexibility test and illustration. T.J.W., N.H, D.Z., J.B., T.Y., T.S., and A.J. analyzed the data. T.J.W. drafted the manuscript, and all authors reviewed and edited the final version.

## Competing interests

The authors declare no competing interest.
