## [Transparent Peer Review file · Nature Communications]

Mechanically flexible mid-wave infrared imagers using black phosphorus ink films

Corresponding Author: Professor Ali Javey

Version 0:

Reviewer comments:

Reviewer #1

(Remarks to the Author)

Single-crystal materials have long been the cornerstone of high-performance detector and imager fabrication due to their advantageous properties: low dark current, high quantum efficiency, and a broad dynamic range enabled by interband transition mechanisms. Alternative mechanisms, such as photogating in devices with disordered interfaces or non-intrinsic band structures, are typically unsuitable for high-quality applications, though they may suffice for systems with less stringent performance requirements. In such cases, cost and long-term stability often become decisive factors.

The authors leverage previously developed BP ink and ink-based infrared detectors [Gupta et al., *Sci. Adv.* 9, eadi9384 (2023)] to fabricate detector arrays on flexible substrates. While the paper's central claim revolves around flexible detectors, the imaging demonstrations (Fig. 2) exclusively use rigid quartz substrates rather than flexible ones. The sole reference to flexible imaging—the “Full-azimuthal imaging” section—lacks experimental imaging results. This omission undermines the work's impact, as the proposed novelty (flexible imagers) remains unsupported by practical demonstrations or application scenarios. Other remarks are as follows

1. **Detectivity over Responsivity:** For practical detector arrays, detectivity (or NEP) is a more critical metric than responsivity, as it accounts for dark current, noise, and device area. While the ink-based detectors may prioritize cost-effectiveness, reporting detectivity values would strengthen the analysis. Supplementary Table 3 should (if possible) include a comprehensive detectivity comparison to contextualize the devices' performance.
2. **Long-Term Stability Concerns:** The long-term stability data in Fig. 3e (hundreds of hours) is insufficient for non-dispensable applications, particularly given the potential fragility of unencapsulated BP films under sustained current flow. Extended testing under operational conditions is possibly necessary to validate durability.
3. **Normalized Data Limitations:** Key figures (e.g., Figs. 3e, 3f, 4d, 5d) present normalized data without absolute values, reducing the paper's informational utility. Including raw data with analysis would enhance transparency and reproducibility.
4. **Mechanistic Ambiguity:** The operational mechanisms of the detectors under biased/unbiased conditions remain unclear. Elucidating these details (e.g., carrier transport, interfacial effects) would deepen the understanding of device physics.
5. **Spectrum Calibration Methodology:** The responsivity calibration method (Lines 155–158) is ambiguously described. The indirect approach needs to be justified. A direct calibration protocol or validation against established standards would strengthen the arguments.

Reviewer #2

(Remarks to the Author)

In this manuscript, the authors present mechanically flexible MWIR imagers based on BP ink films, demonstrating fast response and high responsivity. Leveraging this flexibility, they achieve full-azimuthal imaging, enabling precise detection of directional light sources. Notably, the research group has previously used BP ink films to develop large-scale MWIR emitters and detectors at the single-device level. This work builds on their prior efforts by introducing mechanical flexibility and expanding to device arrays, representing a step toward practical applications. Therefore, I support its publication after the following comments are addressed:

1. The introduction mentions that commercial MWIR detectors based on heteroepitaxial materials face significant challenges, necessitating new materials and device architectures. Given that this manuscript aims to develop MWIR imagers, it is crucial to demonstrate the device response specifically in the MWIR region. Therefore, it is recommended that the authors use an MWIR source instead of the 1550 nm source to characterize the performance shown in Fig. 2. Similarly,

for the imaging results shown in Fig. 3, it would be more appropriate to use an MWIR light source rather than a broadband tungsten halogen source.

2. Fig. 2e shows the transient rise and decay times of a single-pixel BP detector. However, the illumination durations for both the "light on" and "light off" states are limited to about 10 μ s, making it challenging to determine whether the detector can maintain a stable response or if there may be a long-tail slow response caused by trapped states. In addition to the stability, it is recommended to conduct multiple switching cycles to evaluate its repeatability and gain a more comprehensive understanding of the detector's performance.

3. The measured bandwidth of 44 kHz is lower than the RC-limited model value of 88 kHz. The manuscript suggests that this discrepancy indicates additional factors affecting the response time. It is recommended to further describe and discuss these potential factors.

4. In addition to assessing responsivity and response speed, further optoelectronic characterizations are recommended. For the single-device level, it would be beneficial to measure the noise equivalent power (NEP) and detectivity of the detector. For the detector array, evaluating the variation in pixel performance is suggested. Similar characterizations can be found in Fig. 3 of Nature Photonics 11, 366–371 (2017).

5. The manuscript states, "After 17 days, the responsivity remained largely stable, and the difference between the original and final spectra was minimal, indicating the air stability of the encapsulated device (Figure 3d)." However, Fig. 3d does not appear to provide any optical spectra or spectral comparison to support this claim.

Reviewer #3

(Remarks to the Author)

In this paper, authors fabricated flexible mid-wave infrared photodetector arrays based on wafer-scale black phosphorus (BP) ink film. The BP photodetector shows fast response time of several microseconds, reasonable responsivity at room temperature, and good stability after 5000 bending cycles. Authors further shows the application of the flexible detector arrays in panoramic imaging, which shows good performance with an angle prediction error of 2.9 degree. Overall, the results are convincing with sufficient supporting data. The demonstration in the work shows solid evidence that BP can be applied in mid-wave infrared imagers. Hence, I recommend for publication after addressing my minor concerns.

1) Detectivity is a more important factor for characterizing the performance of a photodetector. Can authors provide the measured current noise and detectivity of the photodetector.

2) Can author provide band structure alignment for the ITO/Y6/BP/PEDOT:PSS/TiAu structure in the figure which is beneficial for understanding the electron/hole separation mechanism?

Version 1:

Reviewer comments:

Reviewer #1

(Remarks to the Author)

The authors have revised the paper with more information. Now the analysis is strengthened which supports the claims better. The operational mechanics is clearer. I believe the quality of paper meets the standard of this journal.

Reviewer #2

(Remarks to the Author)

The authors have adequately addressed my comments, and I support its publication.

Reviewer #3

(Remarks to the Author)

In the revised manuscript, authors have improved the manuscript through providing additional information of photodetector performance such as specific detectivity and band alignment of photodetectors. The revised manuscript has met the high standard of Nature Communications. Hence, I recommend for publication.

REPLY LETTER

Manuscript ID: NCOMMS-25-18622

Title: “Mechanically flexible mid-wave infrared imagers using black phosphorus ink films”

Author(s): Theodorus Jonathan Wijaya^{1,2,3,4,†}, Naoki Higashitarumizu^{1,2,3,†}, Shifan Wang^{1,2,3,5}, Shogo Tajima^{1,3,6}, Hyong Min Kim^{1,2,3}, Shu Wang^{2,3,7}, Dehui Zhang^{1,2,3}, James Bullock⁵, Tomoyuki Yokota^{4,*}, Takao Someya^{4,*}, and Ali Javey^{1,2,3,7,8,*}

Email: ajavey@berkeley.edu, someya@ee.t.u-tokyo.ac.jp, yokota@ntech.t.u-tokyo.ac.jp

COMMENTS TO AUTHOR:

Reviewer #1

Single-crystal materials have long been the cornerstone of high-performance detector and imager fabrication due to their advantageous properties: low dark current, high quantum efficiency, and a broad dynamic range enabled by interband transition mechanisms. Alternative mechanisms, such as photogating in devices with disordered interfaces or non-intrinsic band structures, are typically unsuitable for high-quality applications, though they may suffice for systems with less stringent performance requirements. In such cases, cost and long-term stability often become decisive factors.

The authors leverage previously developed BP ink and ink-based infrared detectors [Gupta et al., Sci. Adv. 9, eadi9384 (2023)] to fabricate detector arrays on flexible substrates.

We appreciate the reviewer's thoughtful evaluation of our work. In response to the reviewer's comments, we have revised the manuscript in a point-by-point manner.

Comment #1-1

While the paper's central claim revolves around flexible detectors, the imaging demonstrations (Fig. 2) exclusively use rigid quartz substrates rather than flexible ones. The sole reference to flexible imaging—the “Full-azimuthal imaging” section—lacks experimental imaging results. This omission undermines the work's impact, as the proposed novelty (flexible imagers) remains unsupported by practical demonstrations or application scenarios. Other remarks are as follows.

Reply #1-1

This comment is important. We agree that experimental validation of flexible imaging is essential. Figure 5d presents photocurrent values from the 1×8 flexible imager in terms of photocurrent. To complement this, we now provide spatial photocurrent mapping under varying angular positions of the light source as a practical demonstration of the flexible imaging (Supplementary Figure 15).

Additionally, we performed new measurements using a newly fabricated 6×3 flexible imager, demonstrating multirow–multicolumn imaging, summarized in the new Figure 6. These results expand

the number of tested pixels, enable two-dimensional scanning, and independently confirm the behavior observed in Figure 5d. We further applied localized selective irradiation and successfully reconstructed the corresponding image, validating the imager's spatial resolution capability.

These results highlight the potential of flexible imagers as curved detectors, capable of full-azimuthal imaging, precise detection of directional light sources, as well as flexible 2D scanners.

In response to this comment, we have revised the manuscript. Our revision in the manuscript:

(Main text, line 314-315)

Supplementary Figure 15 presents spatial photocurrent mapping at each angular position of the light source.

(Supplementary Information, Figure 15)

Supplementary Figure 15. Spatial mapping of the photocurrent extracted from the 1×8 flexible imager. Each row corresponds to the measured eight values for a given angular position of the light source (2796 K tungsten halogen lamp, 0.23 W·cm⁻²).

(Main text, line 329-341)

Additionally, multirow–multicolumn imaging was demonstrated using a 6×3-pixel flexible imager conformally attached to a curvilinear surface with a 7.5 mm radius. The six rows spanned a length of 1.8 cm, and the three columns were spaced at 45° intervals. Figure 6a shows an optical photograph of the two-dimensional scanner on the curved surface, while Figure 6b illustrates the device structure and its extended electrode film. Figures 6c and 6d present schematics of the full-field irradiation setup using the same broad-spectrum light source, viewed from the side and front, respectively. In this configuration, the center row was orthogonally irradiated, and all pixels were exposed. All 18 pixels generated photocurrent, with the edge-column pixels detecting $36 \pm 3\%$ of the maximum optical power measured under orthogonal irradiation (Figure 6e). Figures 6f and 6g illustrate the selective irradiation of the scanner, where Rows 3 and 5 received less exposure than Row 4. The corresponding imaging result is shown in Figure 6h, where pixels in Rows 5 and 3 exhibited 61% and 51% of the optical power measured in Row 4, respectively, demonstrating accurate reconstruction of the selective irradiation pattern.

(Main text, Figure 6)

Figure 6. Two-dimensional scanner using a flexible BP imager. (a) Schematic of the flexible imager showing its structure from both sides. The PI film with extended electrode pads is connected to the imager using ACF tape. (b) Optical photograph of a 6×3-pixel flexible imager fabricated on a 25 μm-thick PI film and conformally attached to a curvilinear surface (radius: 7.5 mm). The white dashed line outlines the imager area. (c, d) Schematic illustration of the device under full-field irradiation by a tungsten halogen lamp (2796 K, 0.23 W·cm⁻²), viewed from the side (c) and front (d). Cyan squares indicate the pixel positions. The light source is positioned 1 cm from the device. (e) Imaging result under full-field irradiation, presented as measured optical power per pixel. (f, g) Schematic of the device under selective irradiation, viewed from the side (f) and front (g). (h) Imaging result under selective irradiation, presented as measured optical power per pixel.

Comment #1-2

Detectivity over Responsivity: For practical detector arrays, detectivity (or NEP) is a more critical metric than responsivity, as it accounts for dark current, noise, and device area. While the ink-based

detectors may prioritize cost-effectiveness, reporting detectivity values would strengthen the analysis. Supplementary Table 3 should (if possible) include a comprehensive detectivity comparison to contextualize the devices' performance.

Reply #1-2

We agree. We included the noise equivalent power (Supplementary Figure 5b) and the analysis of specific detectivity D^* . Specifically, we evaluated D^* at 10 Hz across different photon energies (Supplementary Figure 5c). Additionally, relevant D^* values from various devices are listed in Supplementary Table 3 for comparison.

In response to this comment, we have revised the manuscript. Our revision in the manuscript:

(Main text, line 190-192)

Detailed analysis of the noise characteristics and specific detectivity (D^*) of the BP photodetector is provided in Supplementary Note 3 and Supplementary Figure 5.

(Supporting Information, Note 3)

The unbiased device exhibited a noise spectral density (NSD) of $2.3 \times 10^{-13} \text{ A} \cdot \text{Hz}^{-1/2}$ at 10 Hz (Supplementary Figure 5a) and a corresponding noise equivalent power (NEP) of $1.2 \times 10^{-8} \text{ W} \cdot \text{Hz}^{-1/2}$ at $3 \mu\text{m}$ (Supplementary Figure 5b), calculated using $\text{NEP} = \text{NSD}/R$, where R is the responsivity at the specified wavelength. The specific detectivity (D^*) was determined using $D^* = \sqrt{A\Delta f}/\text{NEP}$, where A is the device area and Δf is the integration time (1 s), yielding a detectivity of 4.3×10^6 Jones at $\lambda = 3 \mu\text{m}$. Supplementary Figure 5c shows the variation of D^* with photon energy. Further enhancement of responsivity is required to improve detectivity. To contextualize our device performance, Supplementary Table 3 compares the D^* of our ink-based BP photodetector to that of other reported devices.

(Method, line 449-450)

Noise spectral density was measured using the same lock-in amplifier setup under dark conditions.

(Supplementary Information, Figure 16)

Supplementary Figure 5. Noise characterization and specific detectivity D^* analysis. (a) Noise spectral density (NSD). (b) Noise equivalent power (NEP) evaluated at $\lambda = 3 \mu\text{m}$. (c) Specific detectivity D^* as a function of photon energy at 10 Hz.

(Supplementary Information, Table 3)

Supplementary Table 3. Specifications of MWIR imagers compared to those in this study.

Ref.	Material	MWIR range (μm)	Flexibility	Unbiased operation	Uncooled operation	D^* (Jones)	Rise time	Note
This study	BP	3–4	✓	✓	✓	4.3×10^6 ($\lambda = 3.0 \mu\text{m}$)	4 μs	Demonstrated scalability
Other imager reports								
G. Mu et al. , (2024) ⁶ .	PbS/HgTe	3–5	–	×	✓	3.9×10^9 (1500 °C source)	350 μs	Toxic compound (Hg, Pb)
T. Xu et al. , (2024) ⁷ .	BP/MoS ₂	3–4	–	✓	✓	1.1×10^9 ($\lambda = 3.0 \mu\text{m}$)	88 μs	Limited scalability (manual exfoliation)
Commercial imager products								
FLIR A6780	InSb	3–5	×	×	×			Thermal camera
Hamamatsu P5968-060	InSb	3–5	×	✓	(77 K operation)	1×10^{11} ($\lambda_{\text{peak}} = 3.0 \mu\text{m}$, 77 K) 1.6×10^{11} ($\lambda_{\text{peak}} = 5.3 \mu\text{m}$, 77 K)	30 ns	
Hamamatsu P16702-011MN	InAsSb	3–8	×	✓	✓	1.6×10^6 ($\lambda = 3.0 \mu\text{m}$) 8.8×10^6 ($\lambda_{\text{peak}} = 7.4 \mu\text{m}$)	4 ns	Toxic compound (As)
Thorlabs VML8T0	HgCdTe	3–8	×	✓	✓	3.5×10^8 ($\lambda = 3.0 \mu\text{m}$) 6.0×10^6 ($\lambda_{\text{peak}} = 6.3 \mu\text{m}$)	4 ns	Toxic compound (Hg, Cd)
Hamamatsu P16309-01	(QCIP)	~3.5–6	×	✓	✓	1.5×10^9 ($\lambda_{\text{peak}} = 4.65 \mu\text{m}$)	2 ps	

Explanatory note: The table presents only the reported data, with unreported features indicated by '–'. The MWIR range shows the intersection between reported data and the whole MWIR range ($\lambda = 3–8 \mu\text{m}$). Some studies explicitly identify the photoconductive mechanism for photodetection, excluding

operation at 0 V, while others do not mention operation at 0 V. Some detectivity values for commercial products were calculated from specification sheets, using data at their peak wavelength and at $\lambda = 3.0 \mu\text{m}$ at room temperature, unless otherwise noted. The rise time of commercial products is estimated by their RC-limited response, $t_{\text{rise}} \approx 0.35/f_{3\text{dB}}$, where $f_{3\text{dB}}$ is the reported cutoff frequency. QCIP: quantum cascading infrared photodetectors.

Comment #1-3

Long-Term Stability Concerns: The long-term stability data in Fig. 3e (hundreds of hours) is insufficient for non-dispensable applications, particularly given the potential fragility of unencapsulated BP films under sustained current flow. Extended testing under operational conditions is possibly necessary to validate durability.

Reply #1-3

We agree. We have extended the air-storage stability test to 3600 hours, measured following the initial manuscript submission, demonstrating that the encapsulated device maintains stable performance over 150 days (Figure 3e). We also evaluated operational stability under continuous blackbody irradiation at 400 °C ($13 \text{ mW}\cdot\text{cm}^{-2}$) using a newly fabricated device, confirming stable responsivity over 120 hours.

In response to this comment, we have revised the manuscript. Our revision in the manuscript:

(Main text, line 233-237)

The long-term stability of the photodetector was evaluated under both air storage and continuous irradiation conditions. Figure 3e shows the responsivity comparison of encapsulated and non-encapsulated devices. The responsivity of the non-encapsulated device decreased to 48% after 312 hours (13 days), whereas the encapsulated device maintained its full responsivity even after 3600 hours (150 days) of storage. ...

(Main text, line 242-246)

Additionally, the operational stability of the device under constant irradiation was evaluated using a 400 °C blackbody source at a power density of $0.19 \text{ W}\cdot\text{cm}^{-2}$, as shown in Supplementary Figure 12. Under this condition, the encapsulated device maintained its responsivity over 120 hours (5 days), while the responsivity of the non-encapsulated device degraded to 40% of its initial value after 25 hours (~1 day) of continuous exposure.

(Method, line 472-473)

For the operational stability tests, a 400 °C blackbody source ($0.19 \text{ W}\cdot\text{cm}^{-2}$) modulated by a 300 Hz chopper was used.

(Main figure, Figure 3e)

Figure 3. ... (e) Responsivity during the air-storage stability test of the encapsulated and non-encapsulated photodetector.

(Supporting Information, Figure 12)

Supplementary Figure 12. Responsivity during the operational stability test of the encapsulated and non-encapsulated photodetector. Devices were illuminated by a 400 °C blackbody source ($0.19 \text{ W}\cdot\text{cm}^{-2}$) positioned 2 cm from the device.

Comment #1-4

Normalized Data Limitations: Key figures (e.g., Figs. 3e, 3f, 4d, 5d) present normalized data without absolute values, reducing the paper's informational utility. Including raw data with analysis would enhance transparency and reproducibility.

Reply #1-4

We agree. Following the reviewer's comment, we replaced the normalized values with absolute values: responsivity for photodetector characterizations (Figures 3 and 4), photocurrent for imaging involving

non-orthogonal irradiation (Figure 5), or measured power for imaging results (Figures 3 and 6).

In response to this comment, we have revised the manuscript. Our revision in the manuscript:

(Main figures, Figures 3–6)

Figure 3. ... (d) (Top) Optical photograph of the masks, made of quartz/Ti/Au and patterned with the letters “C,” “A,” and “L.” (Bottom) Imaging results of letter patterns, presented as measured optical power per pixel. (e) Responsivity during the air-storage stability test of the encapsulated and non-encapsulated photodetector. (f) Optical spectra of the encapsulated photodetector, before and after the air-storage stability test.

Figure 4. ... (b) Responsivity values measured during the bending test of the BP photodetector fabricated on PI film (thickness: 25 μm). (c) Responsivity values measured during the cyclic bending test of the BP photodetector on PI film (thickness: 125 μm) with a 1-cm bending radius. (d) Optical spectra of the photodetector before and after 5000 bending cycles.

Figure 5. ... (d) Photocurrent of the eight pixels as a function of the angular position of the light source, measured in 15° intervals. Solid lines are visual guides to indicate data trends. The light source is a tungsten halogen lamp (2796 K, $0.23 \text{ W}\cdot\text{cm}^{-2}$). (Inset) Schematic illustration of the light source angle θ , defined with respect to the normal of Pixel 1.

Figure 6. ... (e) Imaging result under full-field irradiation, presented as measured optical power per pixel. ... (h) Imaging result under selective irradiation, presented as measured optical power per pixel.

Comment #1-5

Mechanistic Ambiguity: The operational mechanisms of the detectors under biased/unbiased conditions remain unclear. Elucidating these details (e.g., carrier transport, interfacial effects) would deepen the understanding of device physics.

Reply #1-5

We agree. The reviewer rightly pointed out that we did not clearly elaborate on the transport mechanism and the role of the interface in the photodetector. Planar heterojunction systems composed of BP and various organic electron acceptor molecules are known to form a p–n junction at the interface [1]. In such systems, photogenerated carriers are separated by built-in potential under zero bias, while an applied reverse bias further enhances carrier separation efficiency.

In response to this comment, we have revised the manuscript and included a relevant reference. Our revision in the manuscript:

(Main text, line 153-155)

The I - V characteristics of the photodetector showed distinct differences between dark and illuminated conditions, exhibiting diode-like behavior, with an open-circuit photovoltage and a short-circuit photocurrent (I_{ph}) (Figure 2a).

(Main text, line 178-183)

This behavior is characteristic of a conventional pn-junction photodiode, Supplementary Figure 1 presents the band diagram of the photodiode between the two electrodes with the junction situated at the BP/Y6 interface. Reverse bias enhances the electric field across this interface, facilitating more efficient separation of photogenerated carriers, consistent with observations in similar systems employing planar junctions of solution-processed BP and organic electron acceptors³⁵.

(References)

35 Bai, L. *et al.* Solution-processed black phosphorus/PCBM hybrid heterojunctions for solar cells. *Journal of Materials Chemistry A* **5**, 8280-8286, doi:10.1039/C6TA08140A (2017).

(Supporting Information, Figure 1)

Supplementary Figure 1. Energy level alignment plot of the photodetector under flat band condition between the two electrodes.

[Reference]

[1] Zhang, M., Biesold, G. M., & Lin, Z. A multifunctional 2D black phosphorene-based platform for improved photovoltaics. *Chemical Society Reviews*, **50** (23), 13346-13371 (2021).

Comment #1-6

Spectrum Calibration Methodology: The responsivity calibration method (Lines 155–158) is ambiguously described. The indirect approach needs to be justified. A direct calibration protocol or validation against established standards would strengthen the arguments.

Reply #1-6

We agree. We clarified the spectral calibration methodology, which follows established protocols [2,3]. The FTIR spectrum was first corrected using the internal DTGS detector to account for the source profile, then scaled to match responsivity measured independently at $\lambda = 1.55 \mu\text{m}$ using a laser diode. The laser power was adjusted to approximate the FTIR output, as verified by comparable photocurrent levels. Additionally, we have included Ref. 19, a previous work from our group that describes the calibration procedure in detail, in the Methods section.

In response to this comment, we have revised the manuscript. Our revision in the manuscript:

(Main text, line 158-163)

The measured FTIR spectrum was first normalized to the response of a deuterated triglycine sulfate (DTGS) detector and subsequently scaled to match the responsivity obtained under $\lambda = 1.55 \mu\text{m}$ irradiation using a laser diode. The laser diode delivered an incident power density of $1.6 \text{ W}\cdot\text{cm}^{-2}$, comparable to that of the FTIR source at the same wavelength, as indicated by a similar measured photocurrent. This correspondence enabled accurate scaling of the FTIR spectrum.

(Method, line 437-438)

The FTIR spectrum result was calibrated using a built-in DTGS detector was used to calibrate the relative spectral intensity of the source, as also detailed in Ref.¹⁹.

(References)

19 Bullock, J. *et al.* Polarization-resolved black phosphorus/molybdenum disulfide mid-wave infrared photodiodes with high detectivity at room temperature. *Nature Photonics* **12**, 601-607, doi:10.1038/s41566-018-0239-8 (2018).

[References]

[2] Amani, M., Regan, E., Bullock, J., Ahn, G.H. and Javey, A., Mid-wave infrared photoconductors based on black phosphorus-arsenic alloys. *ACS nano*, **11**(11), pp.11724-11731 (2017).

[3] Bullock, J., Amani, M., Cho, J. *et al.* Polarization-resolved black phosphorus/molybdenum disulfide mid-wave infrared photodiodes with high detectivity at room temperature. *Nature Photon* **12**, 601–607 (2018).

Reviewer #2

In this manuscript, the authors present mechanically flexible MWIR imagers based on BP ink films, demonstrating fast response and high responsivity. Leveraging this flexibility, they achieve full-azimuthal imaging, enabling precise detection of directional light sources. Notably, the research group has previously used BP ink films to develop large-scale MWIR emitters and detectors at the single-device level. This work builds on their prior efforts by introducing mechanical flexibility and expanding to device arrays, representing a step toward practical applications. Therefore, I support its publication after the following comments are addressed:

We appreciate the reviewer's thoughtful evaluation of our work. In response to the reviewer's comments, we have revised the manuscript in a point-by-point manner.

Comment #2-1

The introduction mentions that commercial MWIR detectors based on heteroepitaxial materials face significant challenges, necessitating new materials and device architectures. Given that this

manuscript aims to develop MWIR imagers, it is crucial to demonstrate the device response specifically in the MWIR region. Therefore, it is recommended that the authors use an MWIR source instead of the 1550 nm source to characterize the performance shown in Fig. 2. Similarly, for the imaging results shown in Fig. 3, it would be more appropriate to use an MWIR light source rather than a broadband tungsten halogen source.

Reply #2-1

This comment is important. We agree that demonstrating MWIR photoresponse is essential. While Figure 2b shows the spectral responsivity of the photodetector, confirming MWIR sensitivity corresponding to photon energies between 0.25 eV and 0.41 eV, the reviewer rightly noted that both the single-pixel characterization and imaging demonstrations were conducted using a 1.55 μm laser.

To address this, we performed single-pixel measurements under $\lambda = 3.3 \mu\text{m}$ LED irradiation ($33 \text{ mW}\cdot\text{cm}^{-2}$), as shown in Supplementary Figure 4, using a newly fabricated device. The device exhibited a clear photoresponse, with responsivity deviating by only 14% from the values estimated from spectral and power-dependent data (Figure 2d).

Additionally, pattern imaging performed with a newly fabricated 6×6 array under the same LED source (Supplementary Figure 11) confirms and complements the spectral results, demonstrating reliable MWIR detection and imaging capability.

In response to this comment, we have revised the manuscript. Our revision in the manuscript:

(Main text, line 168-172)

To further validate MWIR sensitivity, the device was irradiated with a $\lambda = 3.3 \mu\text{m}$ LED at a power density of $33 \text{ mW}\cdot\text{cm}^{-2}$, yielding a measured responsivity of $4.2 \mu\text{A}\cdot\text{W}^{-1}$ (Supplementary Figure 4). Based on the spectral characteristics and power dependence (given below), the expected responsivity at $\lambda = 3.3 \mu\text{m}$ and power level was estimated to be $4.8 \mu\text{A}\cdot\text{W}^{-1}$, corresponding to a 14% deviation of the measured value.

(Main text, line 229-231)

The imager was also tested under MWIR irradiation using a $\lambda = 3.3\text{-}\mu\text{m}$ LED. Supplementary Figure 11 presents the corresponding imaging pattern and results, which showed a clear reconstruction with an error percentage of 5.3%.

(Method, line 439-440)

Additionally, an LED with $\lambda = 3.3 \mu\text{m}$ (Hamamatsu L15893-0330ML, $33 \text{ mW}\cdot\text{cm}^{-2}$) was used to confirm the MWIR response.

(Supplementary Information, Figures 4, 11)

Supplementary Figure 4. MWIR photoresponse of the BP PD under irradiation from a $\lambda = 3.3 \mu\text{m}$ LED ($33 \text{ mW}\cdot\text{cm}^{-2}$).

Supplementary Figure 11. Imaging using $\lambda = 3.3\text{-}\mu\text{m}$ LED ($33 \text{ mW}\cdot\text{cm}^{-2}$). (a) Schematic configuration of the imaging pixel array and the shadow mask for MWIR imaging, showing a diamond-shaped exposed region. (b) Captured image revealing a recognizable diamond pattern. The photocurrent of each pixel is normalized to its maximum photocurrent.

Comment #2-2

Fig. 2e shows the transient rise and decay times of a single-pixel BP detector. However, the illumination durations for both the "light on" and "light off" states are limited to about $10 \mu\text{s}$, making it challenging to determine whether the detector can maintain a stable response or if there may be a long-tail slow response caused by trapped states. In addition to the stability, it is recommended to conduct multiple switching cycles to evaluate its repeatability and gain a more comprehensive understanding of the detector's performance.

Reply #2-2

We agree. We included the expanded results of Figure 2e in Supplementary Figure 6, which indicated the absence of a long-tail slow decay in the photodetector.

Additionally, repeated irradiation cycles taken on a newly fabricated device (Supplementary Figure 7) confirmed the reproducibility of the photoresponse.

In response to this comment, we have revised the manuscript. Our revision in the manuscript:

(Main text, line 196-200)

Irradiation duration exceeding the transient response times of the BP photodetector confirmed the absence of a long-tail slow response (Supplementary Figure 6). Additionally, the reproducibility of the photoresponse was verified through consistent behavior across multiple switching cycles (Supplementary Figure 7).

(Supplementary Information, Figures 6-7)

Supplementary Figure 6. Transient rise and fall behavior of the unbiased BP photodetector under $\lambda = 1.55 \mu\text{m}$ laser irradiation ($1.6 \text{ W}\cdot\text{cm}^{-2}$). (a) Full measurement corresponding to Figure 2e. (b) Expanded view of the rising (top) and falling (bottom) segments, demonstrating the absence of long-tail slow components.

Supplementary Figure 7. Repeated switching cycles of an unbiased BP photodetector under $\lambda = 1.55 \mu\text{m}$ laser irradiation ($1.6 \text{ W}\cdot\text{cm}^{-2}$), confirming reproducibility of the photoresponse.

[References]

- [4] Xu, T. *et al.* Van der Waals mid-wavelength infrared detector linear array for room temperature passive imaging. *Science Advances* **10**, eadn0560, doi:doi:10.1126/sciadv.adn0560 (2024).
- [5] Mu, G. *et al.* Visible to mid-wave infrared PbS/HgTe colloidal quantum dot imagers. *Nature Photonics* **18**, 1147-1154, doi:10.1038/s41566-024-01492-1 (2024).

Comment #2-3

The measured bandwidth of 44 kHz is lower than the RC-limited model value of 88 kHz. The manuscript suggests that this discrepancy indicates additional factors affecting the response time. It is recommended to further describe and discuss these potential factors.

Reply #2-3

We agree. The RC-limited model is reported to overestimate the bandwidth of photodetectors made of van-der-Waals layered materials even by three orders of magnitude [6]. To elucidate this point, we elaborated more on the potential factors affecting the discrepancy between the measured bandwidth and the RC-limited model.

In response to this comment, we have revised the manuscript accordingly and included the relevant references. Our revision in the manuscript:

(Main text, line 205-208)

Potential factors include surface defects at the BP/Y6 interface or within Y6 layers⁴³, as well as inefficient charge carrier extraction at the electrode contacts⁴⁴, both of which were reported in analogous systems to suppress high-frequency response⁴⁵.

(References)

- 43 Strauß, F., Schedel, C. & Scheele, M. Edge contacts accelerate the response of MoS₂ photodetectors. *Nanoscale Advances* **5**, 3494-3499, doi:10.1039/D3NA00223C (2023).
- 44 Strauß, F., Zeng, Z., Braun, K. & Scheele, M. Toward Gigahertz Photodetection with Transition Metal Dichalcogenides. *Acc. Chem. Res.* **57**, 1488-1499, doi:10.1021/acs.accounts.4c00088 (2024).
- 45 Jung, H. S., Eun, K., Kim, Y. T., Lee, E. K. & Choa, S.-H. Experimental and numerical investigation of flexibility of ITO electrode for application in flexible electronic devices. *Microsystem Technologies* **23**, 1961-1970 (2017).

[Reference]

- [6] Christine Schedel, Fabian Strauß, & Marcus Scheele. *The Journal of Physical Chemistry C*, **126** (32), 14011-14016, DOI: 10.1021/acs.jpcc.2c04584 (2022).

Comment #2-4

In addition to assessing responsivity and response speed, further optoelectronic characterizations are recommended. For the single-device level, it would be beneficial to measure the noise equivalent power (NEP) and detectivity of the detector. For the detector array, evaluating the variation in pixel performance is suggested. Similar characterizations can be found in Fig. 3 of Nature Photonics 11, 366–371 (2017).

Reply #2-4

We agree. For the single-pixel device, we included the noise equivalent power (Supplementary Figure 5b) and the analysis of specific detectivity D^* . Specifically, D^* was evaluated at 10 Hz across different photon energies (Supplementary Figure 5c).

For the multipixel imager, the variation in pixel performance of the 6×6 array is shown in Supplementary Figure 9, which presents the spatial mapping of responsivity at 0 V and the corresponding histogram. The 36 unbiased pixels exhibited a responsivity of $23 \pm 7 \mu\text{A} \cdot \text{W}^{-1}$, where the standard deviation is approximately 30% relative to the average value.

In response to this comment, we have revised the manuscript. Our revision in the manuscript:

(Main text, line 190-192)

Detailed analysis of the noise characteristics and specific detectivity (D^*) of the BP photodetector is provided in Supplementary Note 3 and Supplementary Figure 5.

(Supporting Information, Note 3)

The unbiased device exhibited a noise spectral density (NSD) of $2.3 \times 10^{-13} \text{ A} \cdot \text{Hz}^{-1/2}$ at 10 Hz (Supplementary Figure 5a) and a corresponding noise equivalent power (NEP) of $1.2 \times 10^{-8} \text{ W} \cdot \text{Hz}^{-1/2}$ at $3 \mu\text{m}$ (Supplementary Figure 5b), calculated using $\text{NEP} = \text{NSD}/R$, where R is the responsivity at the specified wavelength. The specific detectivity (D^*) was determined using $D^* = \sqrt{A\Delta f}/\text{NEP}$, where A is the device area and Δf is the integration time (1 s), yielding a detectivity of 4.3×10^6 Jones at $\lambda = 3 \mu\text{m}$. Supplementary Figure 5c shows the variation of D^* with photon energy. Further enhancement of responsivity is required to improve detectivity. To contextualize our device performance, Supplementary Table 3 compares the D^* of our ink-based BP photodetector to that of other reported devices.

(Method, line 449-450)

Noise spectral density was measured using the same lock-in amplifier setup under dark conditions.

(Main text, line 219-221)

The 36 unbiased pixels exhibited a responsivity of $23 \pm 7 \mu\text{A} \cdot \text{W}^{-1}$ under $\lambda = 1.55 \mu\text{m}$ irradiation. Supplementary Figure 9 presents the spatial mapping of responsivity across the imager pixels and the corresponding frequency distribution.

(Supplementary Information. Figures 5, 9)

Supplementary Figure 5. Noise characterization and specific detectivity D^* analysis. (a) Noise spectral density (NSD). (b) Noise equivalent power (NEP) evaluated at $\lambda = 3 \mu\text{m}$. (c) Specific detectivity D^* as a function of photon energy at 10 Hz.

Supplementary Figure 9. Variation analysis of responsivity of the imager. (a) Spatial mapping of responsivity at 0 V under $\lambda = 1.55 \mu\text{m}$ irradiation ($1.6 \text{ W}\cdot\text{cm}^{-2}$), measured across 36 pixels. (b) Histogram of the extracted responsivity values.

Comment #2-5

The manuscript states, “After 17 days, the responsivity remained largely stable, and the difference between the original and final spectra was minimal, indicating the air stability of the encapsulated device (Figure 3d).” However, Fig. 3d does not appear to provide any optical spectra or spectral comparison to support this claim.

Reply #2-5

We agree. The reviewer correctly pointed out the typographical error in our text. The storage period should be “14 days,” and the correct subfigure panel is Figure 3f. We have carefully reviewed the manuscript to eliminate typographical errors and ensure the accuracy of the revised version.

In response to this comment, we have revised the manuscript. Our revision in the manuscript:

(Main text, line 237-240)

The spectral difference between the initial measurement and that after 336 hours (14 days) was minimal, confirming the spectral stability of the encapsulated device (Figure 3f). Minor spectral variations were attributed to slight differences in sample positioning during measurement.

(Main text, Figure 3f)

Figure 3. ... (f) Optical spectra of the encapsulated photodetector, before and after the air-storage stability test.

Reviewer #3

In this paper, authors fabricated flexible mid-wave infrared photodetector arrays based on wafer-scale black phosphorus (BP) ink film. The BP photodetector shows fast response time of several microseconds, reasonable responsivity at room temperature, and good stability after 5000 bending cycles. Authors further shows the application of the flexible detector arrays in panoramic imaging, which shows good performance with an angle prediction error of 2.9 degree. Overall, the results are convincing with sufficient supporting data. The demonstration in the work shows solid evidence that BP can be applied in mid-wave infrared imagers. Hence, I recommend for publication after addressing my minor concerns.

We appreciate the reviewer for the thoughtful evaluation of our work. By following the reviewer's comments, we revised the manuscript in a point-by-point manner.

Comment #3-1

Detectivity is a more important factor for characterizing the performance of a photodetector. Can authors provide the measured current noise and detectivity of the photodetector.

Reply #3-1

We agree. We included the noise spectral density (Supplementary Figure 5a) and the analysis of specific detectivity D^* . Specifically, we evaluated D^* at 10 Hz across different photon energies (Supplementary Figure 5c).

In response to this comment, we have revised the manuscript. Our revision in the manuscript:

(Main text, line 190-192)

Detailed analysis of the noise characteristics and specific detectivity (D^*) of the BP photodetector is

provided in Supplementary Note 3 and Supplementary Figure 5.

(Supporting Information, Note 3)

The unbiased device exhibited a noise spectral density (NSD) of $2.3 \times 10^{-13} \text{ A} \cdot \text{Hz}^{-1/2}$ at 10 Hz (Supplementary Figure 5a) and a corresponding noise equivalent power (NEP) of $1.2 \times 10^{-8} \text{ W} \cdot \text{Hz}^{-1/2}$ at $3 \mu\text{m}$ (Supplementary Figure 5b), calculated using $\text{NEP} = \text{NSD}/R$, where R is the responsivity at the specified wavelength. The specific detectivity (D^*) was determined using $D^* = \sqrt{A\Delta f}/\text{NEP}$, where A is the device area and Δf is the integration time (1 s), yielding a detectivity of 4.3×10^6 Jones at $\lambda = 3 \mu\text{m}$. Supplementary Figure 5c shows the variation of D^* with photon energy. Further enhancement of responsivity is required to improve detectivity. To contextualize our device performance, Supplementary Table 3 compares the D^* of our ink-based BP photodetector to that of other reported devices.

(Method, line 449-450)

Noise spectral density was measured using the same lock-in amplifier setup under dark conditions.

(Main text, line 219-221)

The 36 unbiased pixels exhibited a responsivity of $23 \pm 7 \mu\text{A} \cdot \text{W}^{-1}$ under $\lambda = 1.55 \mu\text{m}$ irradiation. Supplementary Figure 9 presents the spatial mapping of responsivity across the imager pixels and the corresponding frequency distribution.

(Supplementary Information, Figure 5)

Supplementary Figure 5. Noise characterization and specific detectivity D^* analysis. (a) Noise spectral density (NSD). (b) Noise equivalent power (NEP) evaluated at $\lambda = 3 \mu\text{m}$. (c) Specific detectivity D^* as a function of photon energy at 10 Hz.

Comment #3-2

Can author provide band structure alignment for the ITO/Y6/BP/PEDOT:PSS/TiAu structure in the figure which is beneficial for understanding the electron/hole separation mechanism?

Reply #3-2

We agree. In response to this comment, we have revised the manuscript and included a relevant reference. Our revision in the manuscript:

(Main text, line 178-183)

This behavior is characteristic of a conventional pn-junction photodiode, Supplementary Figure 1 presents the band diagram of the photodiode between the two electrodes with the junction situated at the BP/Y6 interface. Reverse bias enhances the electric field across this interface, facilitating more efficient separation of photogenerated carriers, consistent with observations in similar systems employing planar junctions of solution-processed BP and organic electron acceptors³⁵.

(References)

[35] Bai, L. *et al.* Solution-processed black phosphorus/PCBM hybrid heterojunctions for solar cells. *Journal of Materials Chemistry A* **5**, 8280-8286, doi:10.1039/C6TA08140A (2017).

(Supporting Information, Figure 1)

Supplementary Figure 1. Energy level alignment plot of the photodetector under flat band condition between the two electrodes.